# Multimodal Task Vectors Enable Many-Shot Multimodal In-Context Learning

**Brandon Huang**[1*]     **Chancharik Mitra**[1*]     **Assaf Arbelle**[2]     **Leonid Karlinsky**[3]

**Trevor Darrell**[1]     **Roei Herzig**[1,2]

[1] University of California, Berkeley     [2] IBM Research     [3] MIT-IBM Watson AI Lab

## Abstract

The recent success of interleaved Large Multimodal Models (LMMs) in few-shot learning suggests that in-context learning (ICL) with many examples can be promising for learning new tasks. However, this *many-shot* multimodal ICL setting has one crucial problem: it is fundamentally limited by the model's context length set at pretraining. The problem is especially prominent in the multimodal domain, which processes both text and images, requiring additional tokens. This motivates the need for a multimodal method to compress many shots into fewer tokens without finetuning. In this work, we enable LMMs to perform multimodal, many-shot in-context learning by leveraging Multimodal Task Vectors (MTV)—compact implicit representations of in-context examples compressed in the model's attention heads. Specifically, we first demonstrate the existence of such MTV in LMMs and then leverage these extracted MTV to enable many-shot in-context learning for various vision-and-language tasks. Our experiments suggest that MTV can scale in performance with the number of compressed shots and generalize to similar out-of-domain tasks without additional context length for inference. Code: https://github.com/Brandon3964/MultiModal-Task-Vector

## 1   Introduction

Large Multimodal Models (LMMs) such as GPT-4V [60], LLaVA [49, 50], and the BLIP [13, 43] family of models demonstrate state-of-the-art performance on a variety of vision and language (VL) tasks due to their strong reasoning capabilities over both text and images. Recent works show that LMMs pre-trained on interleaved text-image data can do multimodal in-context learning [6, 39]. In particular, few-shot, in-context learning (ICL) in text-only LLMs has been scaled with an increasing number of examples in long-context language models—a setting called many-shot learning [1]. A natural question arises on how to perform many-shot learning in the multimodal domain.

The first issue with directly applying a many-shot learning regimen to LMMs is the intrinsic limitation of context length. This is especially true in the multimodal domain, as LMMs must encode both text and images, whose embeddings are token-expensive. Moreover, long-context language models, which LMMs leverage for reasoning, struggle to use their entire context length effectively for ICL [45, 51]. Secondly, perhaps due to the misalignment of pretraining tasks with ICL, many instruction-tuned LMMs underperform on tasks in the ICL setting [16], suggesting the importance of interleaved LMMs. Finally, there is also the challenge of the increasing memory and run-time required for processing long contexts for every inference call. These challenges motivate a method for compressing multimodal in-context examples into compact, implicit representations. Therefore, in this paper, we propose

---

*Denotes Equal Contribution

38th Conference on Neural Information Processing Systems (NeurIPS 2024).

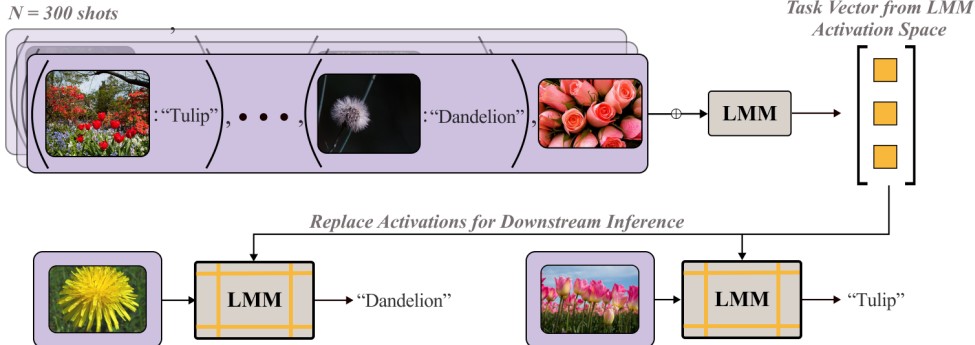

Figure 1: **Multimodal Task Vectors (MTV) Overview.** We overcome an LMM's context length limitation by encoding many shots of multimodal examples as activations in the LMM's latent space. We then directly replace this encoding into the LMM's activation space during downstream inference.

Multimodal Task Vectors (MTV)—compact representations of multimodal in-context tasks—within the attention heads of LMMs to enable many-shot ICL. In particular, we show the existence of MTV in interleaved LMMs, and we use them to compress large numbers of multimodal ICL examples.

Recent research in explainability has demonstrated the existence of task vectors in both the language [25, 81] and vision [27] domains. These task vectors are implicit representations of in-context tasks represented by sets of activations in the model. These activations compactly summarize the information in ICL examples. In our work, we go beyond proving the existence of these task vectors in the multimodal domain by demonstrating their ability to compress examples for many-shot ICL in LMMs without the need for finetuning.

Our method can be described in three steps. First, given a set of many-shot multimodal ICL examples, we calculate the mean activations corresponding to the last token across multiple inference iterations. Second, to avoid the context length constraint, we select a set of attention heads in the model to store the mean activations of the ICL examples. However, since the downstream task may be zero-shot or use a different number of ICL examples, we select a set of examples aligned with its form. We then use these examples to find an optimal set of LMM head locations where the many-shot examples will be encoded. We refer to these mean activations and locations as MTV, which implicitly encodes the many-shot multimodal examples for use in the downstream task. Finally, for downstream inference, we replace the mean activations from Step 1 with the attention head locations found in Step 2. Since we input examples to the LMM across different iterations in Step 1, Multimodal Task Vectors can implicitly encode more examples than are allowable by the context limit. We find that utilizing many examples for extracting MTV surpasses performance on zero-shot and most standard few-shot ICL settings, suggesting the effectiveness of our method. Another key benefit of our method is that it frees up tokens for the model during downstream inference compared to standard few-shot ICL methods. An overview of our method is shown in Figure 1.

We summarize our main contributions as follows: (i) We show the existence of Multimodal Task Vectors, compact implicit representations of in-context functions in LMMs. (ii) MTV can encode more examples than allowed by an LMM's context length, enabling both runtime and memory-efficient multimodal many-shot in-context learning. (iii) MTV surpasses zero-shot and few-shot ICL settings on various VL benchmarks without finetuning. (iv) MTV can scale to larger numbers of examples and can generalize to similar out-of-domain tasks.

## 2 Related Works

**Many-Shot In-Context Learning**. Few-shot in-context learning (ICL) is a significant area of study in text-only LLMs [9, 89]. A natural question arises about the possibility of using a larger number of shots (e.g., hundreds) to further improve performance or learn more complex tasks. Indeed, some early work in text-only *many-shot, in-context learning* suggests performance on different tasks can scale with a larger number of examples [1, 7, 44, 45].

However, scaling ICL in text-only LLMs is a challenge due to the intrinsic context length. One method to increase context length in these models is to apply positional interpolation methods [10, 63]. However, research on these longer-context models finds that they struggle to use the entire context for ICL [45, 51]. Moreover, as inference on long contexts of inputs is also time and memory-expensive, it is unclear whether simply scaling the context of models is practical for enabling multimodal many-shot ICL in open-source models. There is some early evidence of multimodal many-shot ICL being effective in closed-source models [35], so the question arises as to how to achieve something similar for open-source models. This has led to work that looks to compress explicit input tokens [11, 20, 34, 58, 73, 77]. But crucially, many of these methods require finetuning and only try to preserve performance. Our work is different in that it is the first to enable *multimodal* models with many-shot ICL capabilities, while also improving on complex VL tasks without finetuning.

**Task Vectors**. Our work builds off of research in text-only and vision-only domains showing that internal representations of these models called task vectors [25, 27, 81] (or function vectors) can encapsulate tasks outlined by ICL examples. Our is the first demonstration of Multimodal Task Vectors (MTV) in LMMs. Going beyond previous work, however, we show that MTV enable LMMs not only to use many-shot, multimodal ICL examples but also scale with more samples, be used alongside explicit ICL shots, and even generalize to unseen classes or similar tasks.

**Model Domain Adaptation Methods**. As LLM and LMM model architectures have advanced, so have methods to allow these models to generalize beyond their pretraining distributions. Methods like instruction tuning [5, 50, 68, 88] have shown strong zero-shot generalization to some out-of-domain tasks, but forgetting remains an issue. One popular solution to this issue involves Parameter Efficient Fine-tuning (PEFT) [28]: finetuning either a set of soft prompt input tokens [41, 46], low-rank model weights [14, 29, 99], or a separate adapter from the main model [18, 30, 100].

Prompting methods are a well-explored area for adapting models without finetuning. LLM prompting includes zero-shot methods [36, 83, 85], few-shot and ICL methods [9, 15, 53, 56], expert prompting [93], and Chain-of-Thought (CoT) [90, 101], with extensions like self-consistency [86], Tree-of-Thought (ToT) [94], and Graph-of-Thought (GoT) [8, 40, 95] for more complex structures. Similar multimodal prompting methods exist for LMMs as well [57, 84, 87, 102, 104].

**Large Multimodal Models (LMMs)**. The state-of-the-art performance of LMMs [2, 6, 13, 17, 21, 43, 49, 50, 96, 97, 105] on multimodal tasks stems from combining LLMs' reasoning capabilities [3, 12, 26, 66, 71, 78] with the perception abilities of vision models. LMMs' generative reasoning also makes them more applicable to complex tasks than previous contrastive methods [42, 43, 65]. Such tasks include visual question-answering [4, 23, 24, 31, 32, 54, 67] as well as object identification and localization [37, 48, 59, 82]. Visual Programmatic Models (VPMs) are another class of multimodal methods that makes use of in-context APIs code generation [19, 22, 52, 64, 69, 72, 74, 76, 92]. However, context length limits both LMMs' and VPMs' ability to use multimodal prompting methods such as ICL [9]. Another key challenge is that many LMMs are pre-trained on single text-image pair data. Recently, many LMM models now pretrain on interleaved text-image data [2, 6, 16, 33, 39, 75, 103], making effective multimodal ICL possible. In our work, MTV goes beyond simple few-shot multimodal ICL and scales to many-shot multimodal ICL.

## 3 Multimodal Task Vectors

To address the challenge of performing many-shot multimodal in-context learning, we demonstrate the existence of MTV in LMMs and then leverage them for many-shot multimodal ICL. We begin by describing some background on multimodal ICL and task vectors (Section 3.1). We then introduce our three-step approach: (i) We calculate the mean activations of the attention heads from the many-shot multimodal ICL examples (Section 3.2); (ii) We then extract the set of LMM attention heads locations that best align to the downstream task using an adapted version of the REINFORCE [91] algorithm (Section 3.3); and (iii) We replace the calculated mean activation values into the LMM for a downstream task (Section 3.4). The detailed method visual is shown in Figure 1.

### 3.1 Preliminaries

In the multimodal in-context learning setting, an LMM learns a new task outlined by a set of multimodal examples. The input to the LMM would be outlined as follows:

$$I_{\text{few}} = [(x_1 : y_1), (x_2 : y_2), \ldots, (x_n : y_n), Q] \tag{1}$$

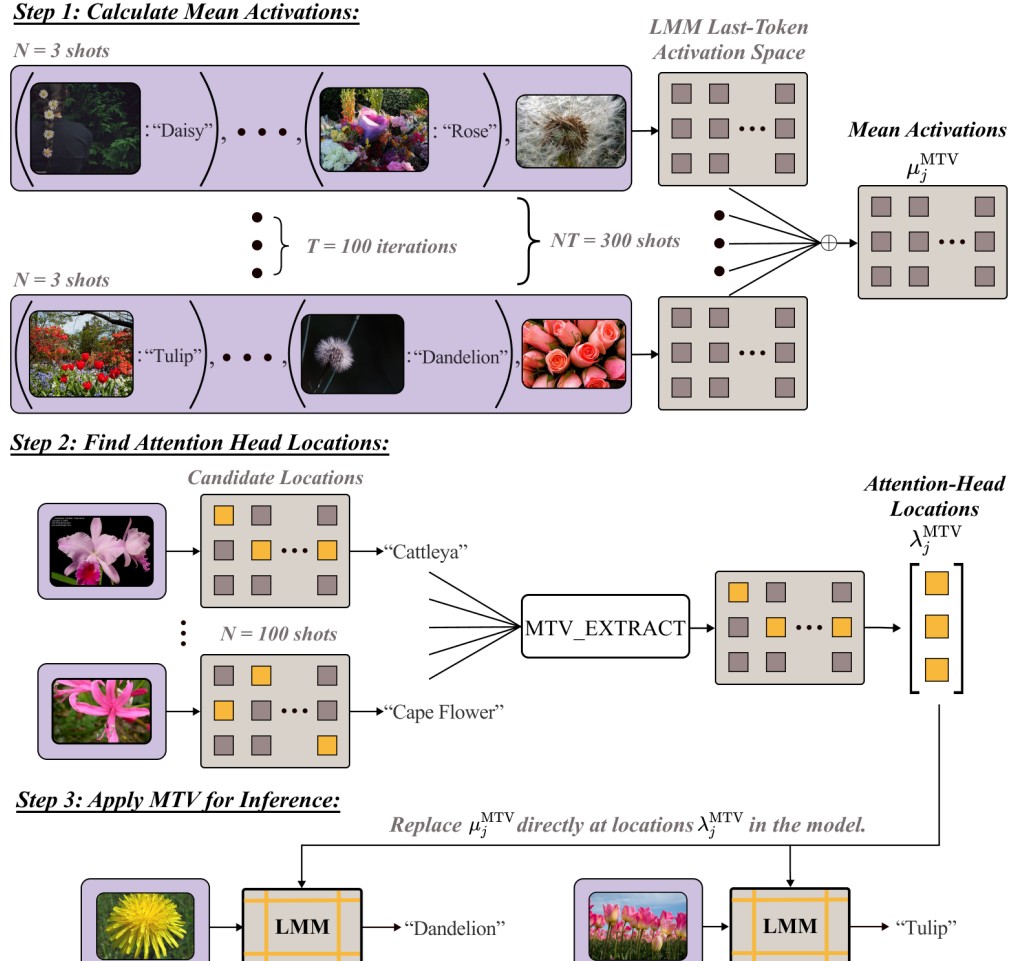

Figure 2: **Multimodal Task Vectors (MTV).** In the standard multimodal in-context learning (ICL) paradigm, the number of shots is limited by an LMM's context length. We solve this issue by first finding the mean activations corresponding to the last token of the examples' input (Step 1), and then calculating a set of attention head locations (Step 2) that best align with the downstream task. These mean activations are then replaced directly in these attention head locations (Step 3), enabling many-shot multimodal ICL.

where the model is prompted to answer a query $Q$ given a set of input-output examples (each $x_i$ being a multimodal input and each $y_i$ a text output).

We note that in-context examples are commonly passed sequentially to the LMM, necessarily restricting multimodal ICL to being small numbers of shots due to limited context length. Furthermore, the images require more tokens to embed, which means enabling many-shot ICL is even more challenging in the multimodal domain. To solve this, we utilize our method MTV—which are implicit representations in the model's attention heads that encode a many-shot multimodal ICL task.

We start with a background on task vectors for some task $j$. Given a model $F$, we denote the set of attention-head locations as $\lambda = \{l \mid \forall l \in F\}$ where each location $l$ is indexed as $l = (h, m)$ for the $h^{\text{th}}$ layer and $m^{\text{th}}$ attention head. Now, task vectors utilize the intermediate outputs of an LMM, called **activations**. For a given input sequence of written in terms of its tokens $x = \{x_1, x_2, \ldots, x_T\}$, each attention head $(h, m)$ produces an activation $z_l \in \mathbb{R}^{\frac{d}{H}}$ for each token $x_i$, where $d$ is the model's embedding dimension and $H$ is the number of heads. These activations are simply the output vectors of each attention head *before* any linear projection. While each head's activation is typically concatenated with others and projected to form the layer's output, task vectors specifically utilize the

pre-projection activations of the final token $x_T$ from each attention head. We thus define the task vectors as follows: (1) the task vector **values** $\mu_j$ are a subset of mean activations produced by the attention heads of $F$ given examples of a task, and (2) the task vector **locations** $\lambda_j$, which denotes a subset of the attention head indices per task. Thus, the task vector is $(\mu_j, \lambda_j)$. For inference, $\mu_j$ replaces the activation values of the heads in the locations given by $\lambda_j$.

In prior work [25, 27, 81], the calculation of the mean activations $\mu_j$ and the extraction of the attention-head locations $\lambda_j$ are used together to extract the task vector. Interestingly, we find that these two steps should be decoupled in order to better align with the downstream task. In our work, we calculate the mean activations $\mu_j$ corresponding to the last token specifically to encode a dataset of many-shot multimodal ICL examples by averaging them across multiple inference calls. However, the downstream task may not always be in the same ICL format as the many-shot examples (e.g., the downstream task uses a different number of shots or is zero-shot). To solve this, we use a separate set of examples that are of the exact format of the downstream task to align the extracted attention-head locations $\lambda_j$ with the inference task. This separation of responsibilities, wherein $\mu_j$ captures the essential information from the many-shot examples and $\lambda_j$ identifies the specific attention head locations for the downstream task, optimizes the utilization of the encoded information at relevant locations within the model.

Our approach to finding Multimodal Task Vectors (MTV) $(\mu_j^{\text{MTV}}, \lambda_j^{\text{MTV}})$ allows LMMs to actually leverage many-shot multimodal ICL examples for complex vision-language tasks without being limited by context length. We proceed by first describing how to calculate the mean activations.

## 3.2 Step 1: Calculate MTV Mean Activations

The ultimate objective of many-shot multimodal ICL is to use a large number of input-output examples when solving a task $j$. However, it is not trivial to get the LMM to see more examples during inference time than its context length allows.

To address this issue, we pass a few-shot input $I_t$ for each inference call $t$ for a total of $T > 1$ inference calls. Each $I_t$ consists of $N$ shots (where $N > 1$) of multimodal in-context examples in the form of randomly-selected input-output response pairs $(x_t : y_t)$, and $Q_t$, which is the query to be answered by the LMM in that iteration.

$$I_t = [(x_1 : y_1), (x_2 : y_2), \ldots, (x_N : y_N), Q_t] \tag{2}$$

Thus, over $T$ LMM inference calls, we have a many-shot multimodal dataset (of $N \times T$ examples):

$$I_{\text{many}} = [I_1, I_2, \ldots, I_T] \tag{3}$$

However, this dataset is still just a disconnected set of few-shot examples. Next, we would like to connect the separate examples into one unified many-shot multimodal ICL representation.

For each inference call, the LMM is given $N$-shot ICL examples. We calculate the mean of the activations corresponding to the last token of the input $z_{l,j}$ for each attention head index $\forall l \in \lambda$ (Section 3.1) across $T$ inference calls, yielding:

$$\forall l \in \lambda: \quad \mu_{l,j} = \frac{1}{T} \sum_{t=1}^{T} \mathbb{E}[z_{l,j} \mid I_t] = \frac{1}{T} \sum_{t=1}^{T} \mathbb{E}\left[z_{l,j} \mid (x_1 : y_1), (x_2 : y_2), \ldots, (x_N : y_N), Q_t\right] \tag{4}$$

In this step, we have found the mean activations $\mu_{l,j}$, which encode an internal LMM representation of many shots of multimodal ICL examples. In the next subsection, we describe our methodology for selecting the set of attention heads where these mean activations will be used.

## 3.3 Step 2: Extract MTV Attention Head Locations

After Step 1, we now have mean activations for the attention heads of the last token in a given many-shot multimodal task. Yet, we still need to find which set of attention heads $\lambda_j^{\text{MTV}}$ should be chosen to encode our task.

To choose the set of attention heads, we first prepare a separate set of $S$ examples specifically aligned to the format of the downstream task. For instance, if the downstream setting is a 2-way, one-shot

classification task, then the $S$ examples should conform to this paradigm. For our explanation, let's consider a downstream task that is zero-shot such that there is a single query $Q_s$ and corresponding response $R_s$ for all $s \in [1, 2, \ldots, S]$.

From these examples, we utilize an adapted version of the REINFORCE [91] algorithm—an iterative policy optimization method that can be used to find task vector locations [27]. Given an LMM $F$, we first select a proposed set of attention head locations by sampling a Bernoulli distribution over the locations multiple times. Next, we directly replace the values of the selected attention heads with the corresponding mean activations $\mu_{l,j}$. Then, after prompting the model with the query $Q_s$, we use the negative cross-entropy loss between the LMM's output logits and the logits of the ground-truth response $R_s$ to optimize the Bernoulli distribution. By optimizing the Bernoulli distribution across $S$ iterations, we are finding the best attention head locations $\lambda_j^{\text{MTV}}$ for patching in our mean activations. Finally, we can extract $\lambda_j^{\text{MTV}}$, the optimized indices of attention heads, by sampling our optimized Bernoulli distribution.

$$\lambda_j^{\text{MTV}} = \text{MTV\_EXTRACT}(F, [Q_1, Q_2, \ldots, Q_S)], [R_1, R_2, \ldots, R_S]) \tag{5}$$

It is important to note that MTV_EXTRACT does not require finetuning of the LMM parameters, but rather only inference calls. We describe further the underlying details of our adapted MTV_EXTRACT algorithm in Section A.2 of the Supplementary. Having found $\lambda_j^{\text{MTV}}$ and $\mu_{l,j}$, we describe in what follows, the final procedure to use MTV for inference.

### 3.4 Step 3: Multimodal Task Vector Application

After we have identified a set of attention heads $\lambda_j^{\text{MTV}}$, it is straightforward to apply MTV for inference. We denote the set of mean activations $\mu_j^{\text{MTV}}$ as follows $\mu_j^{\text{MTV}} = \{\mu_{l,j} | \forall l \in \lambda_j^{\text{MTV}}\}$.

To run downstream inference on a new query $Q_{\text{new}}$ with our model $F$, we directly replace the values of attention heads $\lambda_j^{\text{MTV}}$ with $\mu_j^{\text{MTV}}$ and produce the following response $R_{\text{new}}$:

$$R_{\text{new}} = F(Q_{new} | \lambda_j^{\text{MTV}}, \mu_j^{\text{MTV}}) \tag{6}$$

$R_{\text{new}}$ is thus a response generated using many shots of multimodal examples as implicit context via MTV. The key insight of our method is the importance of $N$ (the number of multimodal examples) and many $T$ (the number of iterations) during the calculation of MTV. This enables an LMM to go beyond its context length to learn more nuanced properties of the task from seeing many examples. Additionally, insertion of MTV directly into the LMM also obviates the need for any context length during downstream inference, actually *freeing* additional context for other use (e.g., an additional prompt, more ICL examples, etc.). Finally, because we align the attention-head locations with the downstream task, MTV can be effectively applied to zero-shot and different ICL settings.

## 4 Evaluation

In order for LMMs to perform multimodal ICL, it is important for interleaved data to be included in pretraining. We apply our MTV approach to Qwen-VL [6], Idefics2-8B [38], and ViLA-1.5-8B [47] three popular interleaved LMMs. For each model, we compare our method to using few-shot ICL across different vision-and-language tasks like VQA and object identification.

### 4.1 Implementation Details

We implemented MTV using PyTorch [61]. We used each model's respective official implementation. While the compute and memory requirements differ slightly between models, all our experiments can be run on a single NVIDIA A6000 GPU. For additional information, refer to Supplementary Section B. Our model and weights will be released upon acceptance, and our code is in Supplementary.

### 4.2 Models

In this work, we apply MTV to the following interleaved LMMs as they are better-suited for multimodal ICL as shown by [16]: (1) **QwenVL** [6] is a LLaMA-based model that has the ability to

process high-resolution images, and its two-stage pre-training methodology, which includes multi-task finetuning and interleaved text-image data. (2) **Idefics2-8B** [39] is a Mistral-based model that benefits from its pre-training on the expansive OBELICS dataset, which comprises a web-scale collection of interleaved image-text documents. We utilize the base version of the model. This demonstrates multimodal in-context learning abilities. (3) **LLaMA3-ViLA-1.5-8B** (abbreviated as VILA-1.5-8B). ViLA-1.5-8B [47] is an architecture that leverages LLaMA-3 as the LLM backbone. As in others, a significant portion of the model's pretraining data is interleaved text-image data. (4) **MANTIS-LLaMA3-8B**. MANTIS-LLaMA3-8B [33] is a combination of a SigLIP [98] visual encoder and LLaMA3 [55] language model finetuned using the MANTIS dataset, a specially curated multi-image dataset that emphasizes co-reference, reasoning, comparing, temporal understanding.

We show the number of tokens per image embedding for each model in Table 1 to illustrate the especial importance of MTVs in the image-text domain:

Table 1: Per Image Embedding Token Length and Total Context Length for Models

| Model Name | Per Image Token Length | Total Context Length |
|---|---|---|
| VILA-1.5-8B | 144 | 8192 |
| Idefics2-8B | 64 | 8192 |
| QwenVL | 256 | 8192 |
| MANTIS-LLaMA3-8B | 64 | 8192 |

### 4.3 Datasets

We briefly describe the tasks and datasets we evaluate our method on. More details about the datasets and their setup can be found in Section B.

**VQA Datasets**. We use the following commonly-evaluated datasets which emphasize different aspects of multimodal reasoning, including visual features (VizWiz) and outside knowledge (OK-VQA): (1) **VizWiz** [23] consists of images taken by visually impaired individuals paired with questions they pose about these images, making it crucial for developing AI systems that assist in real-world, accessibility-focused visual understanding tasks. (2) **OK-VQA** dataset [54] is designed to push the boundaries of Visual Question Answering (VQA) by focusing on knowledge-based questions, where answers require external knowledge beyond the image content. (3)

**Object Classification**. We use the following datasets, which are commonly used for object classification in multimodal ICL: (1) The **Flowers** dataset [59], commonly known as the Oxford 102 Flowers dataset, is a collection specifically designed for image-based flower species recognition for fine-grained classification of 102 different categories of flowers. (2) **Caltech's CUB Dataset on Birds** [82] is a well-known resource for evaluating algorithms on the task of object identification, specifically focused on bird species. It features 200 bird species with roughly 30 images each, annotated with key attributes and bounding boxes. Both Flowers and Birds are formatted as 2-way,1-shot classification episodes, with model inputs being a positive and negative image for the class to be identified in the query image. The response format is a short text response.

## 5 Results

Our main results are shown in Table 2. For VQA, we show the results of MTV with 4 shots per 100 iterations to calculate the mean activations and 100 examples for task vector locations (500 examples total). The task vector is extracted using examples from the train set of the dataset and evaluated on the validation set. For object classification, we extract MTV based on a 2-way, one-shot regimen per 100 iterations for both mean activations and task vector locations (200 examples total). The task vector is extracted using a train set of 30% of the object classes and evaluated on the remaining 70% of *unseen* classes. We demonstrate how Multimodal Task Vectors outperforms zero-shot and few-shot ICL settings on three different models on VL tasks, highlighting the effectiveness of our method. Next, we describe the unique capabilities of our method, such as scaling to more samples and showing some generalizations to other tasks. More results can be found in Section A.1 of Supplementary.

Table 2: **Results**. (Left) MTV evaluated on VQA datasets. (Right) MTV evaluated on object classification datasets. The baselines are shown in gray.

(a) **MTV on VQA Benchmarks**

| Model | VizWiz | OK-VQA |
|---|---|---|
| Flamingo 9B | 28.8 | 44.7 |
| +4-shot ICL | 34.9 | 49.3 |
| +8-shot ICL | 39.4 | 50.0 |
| Blip3 | 21.2 | 26.5 |
| +4-shot ICL | 38.4 | 49.2 |
| +8-shot ICL | 44.3 | 49.1 |
| Qwen-VL-7B | 35.2 | 58.6 |
| +4-shot ICL | 42.0 | **62.0** |
| +8-shot ICL | 44.3 | 61.5 |
| **+MTV** | **45.6** | **62.0** |
| Idefics2 | 31.3 | 52.4 |
| +4-shot ICL | 40.8 | 51.5 |
| +8-shot ICL | 43.8 | 52.3 |
| **+MTV** | **52.5** | **53.0** |
| VILA-1.5-8B | 28.0 | 32.8 |
| +4-shot ICL | 39.3 | 35.6 |
| +8-shot ICL | 44.2 | 36.5 |
| **+MTV** | **55.2** | **40.6** |
| MANTIS-LLaMA3-8B | 36.3 | 51.7 |
| +4-shot ICL | 26.4 | 52.5 |
| +8-shot ICL | 27.5 | 52.0 |
| **+MTV** | **51.0** | **52.8** |

(b) **MTV on Object Classification**

| Model | Flowers | CUB |
|---|---|---|
| LLaVA-1.5-13B | | |
| + 1-shot ICL | 58.60 | 58.24 |
| LLaVA-1.6-13B | | |
| + 1-shot ICL | 65.58 | 67.90 |
| Flamingo 9B | | |
| + 1-shot ICL 9B | 48.78 | 51.2 |
| IDEFICS-9B | | |
| + 1-shot ICL | 55.29 | 62.0 |
| Emu 37B | | |
| + 1-shot ICL | 52.76 | 53.56 |
| Qwen-VL-7B | | |
| + 1-shot ICL | 55.0 | 56.5 |
| + **MTV**+1-shot ICL | **78.1** | **80.0** |
| Idefics2 | | |
| + 1-shot ICL | 82.8 | 88.7 |
| + **MTV**+1-shot ICL | **83.8** | **89.8** |
| VILA-1.5-8B | | |
| + 1-shot ICL | 87.4 | 88.4 |
| + **MTV**+1-shot ICL | **89.3** | **89.7** |
| MANTIS-LLaMA3-8B | | |
| + 1-shot ICL | 87.4 | 84.0 |
| + **MTV**+1-shot ICL | **89.8** | **89.7** |

## 5.1 MTV scales with more examples

We are interested in evaluating (i) the effect of different numbers of shots used *per iteration* to extract MTV and (ii) the effect of different numbers of *iterations* used. We test the impact on accuracy when increasing both of these parameters for QwenVL on the VizWiz validation set. In Figure 3, we show on the left that the optimal number of multimodal ICL shots is 16 shots per iteration. Further, we show on the right side of the figure that 1000 examples yield the best performance. These results illustrate that MTV can effectively scale by utilizing larger numbers of ICL examples per iteration and also in aggregate.

## 5.2 MTV works with explicit few-shot examples

One of the benefits of MTV over a few-shot ICL is the context length that is saved during inference. This is because the many-shot examples are encoded directly in the activation space rather than in the input token space. Thus, we ask whether the LMM can use the freed context for additional few-shot examples. For object classification, we formulate both Flowers and CUB as a 1-shot comparison between a positive and negative sample to identify the correct class (i.e., 2-way, 1-shot ICL by construction). We report results on 1-shot ICL and MTV with 1-shot classification during inference. MTV+1-shot ICL surpasses 1-shot ICL accuracy on these tasks, showing that MTV can be utilized alongside few-shot examples. Furthermore, it is vital to note that the evaluation classes are completely unseen by MTV. Thus, with just a 1-shot ICL example, MTV is able to generalize to unseen classes.

## 5.3 MTV heads generalize to other tasks

In this experiment, we further ask whether the MTV heads $\lambda_j^{\mathrm{MTV}}$ extracted on one task $j$ can generalize to a separate, but similar task $k$. To test this, we use the attention heads extracted from

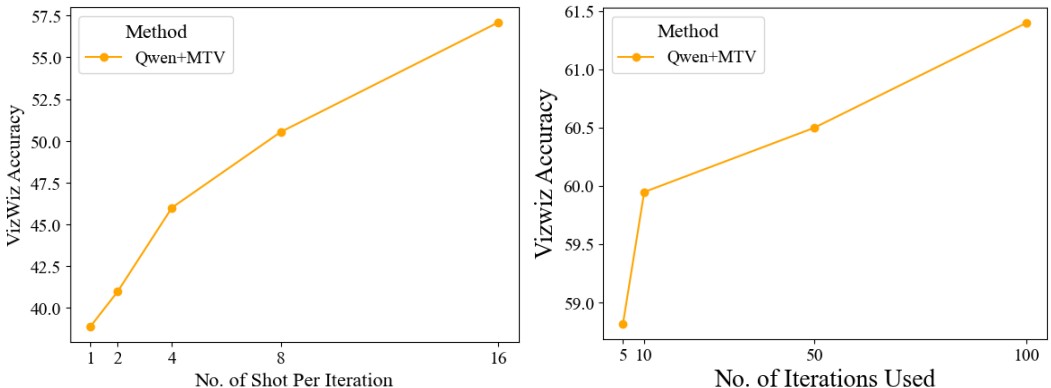

Figure 3: **Scaling of Qwen-MTV on VizWiz:** (Left) We show the effect of varying the number of shots per iteration for a fixed 100 iterations. (Right) We also show the effect of varying numbers of iterations fixing 4 shots per iteration.

ViLA-1.5-8B on VizWiz for use on OK-VQA. Our results on the left of Table 3 demonstrate that the extracted heads from one task can improve accuracy on another similar task. This generalizability of the heads is significant because it suggests that the heads from MTV may only have to be extracted once to be applied to many other similar tasks. The only calculation necessary then would be the mean activations of the many-shot examples used for the target dataset, making the application of many-shot multimodal ICL even more efficient for similar tasks.

Table 3: **Generalization & Method Comparison** (Left) MTV-VizWiz evaluated on OK-VQA. (Right) MTV compared to VizWiz finetuning, function vectors [81], and task vectors [27].

<table>
<tr><td colspan="3">(a) Attention Head Generalization</td><td colspan="3">(b) Comparison to Other Methods</td></tr>
<tr><th>Model</th><th>VizWiz</th><th>OK-VQA</th><th>Model</th><th>VizWiz</th><th>OK-VQA</th></tr>
<tr><td>ViLA-1.5-8B</td><td>28.0</td><td>32.8</td><td>Qwen-VL-7B</td><td>35.2</td><td>58.6</td></tr>
<tr><td>  + 4-shot-ICL</td><td>39.3</td><td>35.6</td><td>  + VizWiz F.T.</td><td>62.0</td><td>25.1</td></tr>
<tr><td>  + 8-shot-ICL</td><td>44.2</td><td>36.5</td><td>  + FV</td><td>36.4</td><td>59.0</td></tr>
<tr><td>  **+ MTV-Vizwiz**</td><td>**55.2**</td><td>**38.3**</td><td>  + VTV</td><td>37.0</td><td>58.9</td></tr>
<tr><td></td><td></td><td></td><td>  **+ MTV**</td><td>**45.6**</td><td>**62.0**</td></tr>
</table>

## 5.4 Finetuning as an upper bound

In Table 3b, we compare our method to finetuning. To do this, we use finetune on the same number of examples as MTV uses from the train set and evaluate not only on the validation set but also on the validation set of another similar dataset. In particular, for a ViLA-1.5-8B model finetuned on VizWiz, we report accuracy on both VizWiz and OK-VQA validation sets. It can be seen that finetuning is indeed an upper bound on the dataset the model was finetuned on. However, we show that finetuning leads to overfitting on the finetuned dataset and even forgetting the zero-shot capabilities. In contrast, we also show that MTV not only improves zero-shot capabilities but can generalize to similar tasks with only a few inference examples Table 2b and Table 3a.

## 5.5 Comparison to other methods

We compare our method to two different methods that can find task vectors: Visual Task Vectors (VTV) [27] and Function Vectors (FV) [81]. Originally, these works could not be applied as-is to support multimodal ICL, but here, we have implemented a version that follows the original exactly with only minor modifications to allow performing our evaluated multimodal tasks. More details about the methods can be found in Section A.2 in the Supplementary. In our experiments Table 3b, we find that MTV surpasses both methods on VizWiz and OK-VQA. VTV are image-only task vectors that use only one-shot image examples for fixed small $T$ iterations, and they calculate the

mean activations and the locations together without aligning to the downstream task. FV are text-only task vectors that use Causal Mediation Analysis [62] to extract task vector locations from only the output activations of the last token. The results suggest the importance of finding the task vectors by decoupling the calculation of the mean activations and locations in two separate steps to perform many-shot multimodal ICL more effectively for complex multimodal tasks.

## 5.6 Compute and runtime efficiency

| Metric | 0-shot | 4-shot | 8-shot | 16-shot | MTV (400-shot) |
|---|---|---|---|---|---|
| Max GPU Memory (GB) | 17.4 | 18.3 | 19.0 | 20.6 | 19.8 |
| Runtime per 100 iterations (min) | 1.1 | 2.7 | 3.1 | 3.3 | 1.9 |

Table 4: **Efficiency:** We show that even though MTV encodes 400 multimodal ICL examples in the mean activations, it still requires less runtime and memory than 8-shot and 16-shot multimodal ICL.

An important feature of our work is that multimodal ICL examples do not require explicit tokens during inference. Because of this, we are interested in the efficiency gains of our method. Intuitively, the longer MTV extraction time is amortized during downstream inference, where the runtime would be equivalent to the zero-shot case. Similarly, the memory requirements are maximal during the MTV extraction process but require the same memory as the zero-shot case afterward. In contrast, the ICL tasks have a slower runtime and larger memory requirement throughout due to running inference on $N$ examples *for every iteration*. To demonstrate this, we calculate the maximum memory requirement in gigabytes (GB) for ViLA-1.5-8B on VizWiz using different ICL-shot counts and MTV with 400 examples. As shown in Table 4, MTV requires less runtime than 16-shot, 8-shot, and 4-shot ICL methods and also requires less memory than 16-shot ICL. These results demonstrate that MTV can encode many multimodal ICL examples with greater efficiency than few-shot methods.

## 6 Conclusion

In this work, we present Multimodal Task Vectors a compact, implicit representation that can efficiently encode many-shot multimodal ICL examples for use in complex vision-language tasks. We demonstrate this implicit model representation not only encodes a multimodal ICL task but can also enable many-shot multimodal ICL to surpass zero-shot and few-shot performance on a variety of VL tasks. Our method stands out from previous work in its ability to scale, use additional explicit multimodal ICL examples, and generalize to other similar VL tasks. Our work is a viable way to surpass the limit of context length of an LMM for multimodal ICL and demonstrates clearly that these additional examples aid in multimodal reasoning. Finally, we do not anticipate a specific negative impact, but, as with any Machine Learning method, we recommend exercising caution.

## 7 Limitations

While Multimodal Task Vectors offers substantial benefits for handling complex vision-language tasks compared to finetuning or few-shot ICL, it is important to recognize certain limitations that accompany our approach. MTV requires access to the internal architecture of an LMM, so while it is an effective solution for all open-source models, its application is restricted from proprietary models, such as GPT-4 [60] and Gemini [79, 80]. Furthermore, while many-shot ICL is incredibly attractive for many applications, it may not be practical for low-data scenarios where synthetic data [1] or the transfer of MTV extracted from another dataset may be required. We feel these challenges represent great opportunities for future work in the many-shot multimodal in-context learning domain.

## 8 Acknowledgements

We would like to thank Deva Ramanan, Grace Luo, and Suzanne Petryk for their insightful feedback and discussions. This project has received funding from Prof. Darrell's group in part by DoD, including PTG and/or LwLL programs, as well as BAIR's industrial alliance programs.

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

# Multimodal Task Vectors Enable Many-Shot Multimodal In-Context Learning

## Supplementary Material

Here, we provide additional information about our experimental results, qualitative examples, implementation details, and datasets. Specifically, Section A provides more experiment results, Section A.2 provides additional method details, Section B provides additional implementation details, and Section C provides qualitative visualizations to illustrate our approach.

## A  Additional Experiment Results

We present several additional experiments that further demonstrate the benefits of our MTV approach.

### A.1  Additional Experiments

Here we provide additional experiments and ablations that further illustrate different characteristics of MTV.

**Motivation for encoding shots in the activation space.**. We highlight our paper's motivation in addressing the context length token limitation of LMMs by encoding ICL shots in the activation space. An additional limiting factor in the token space is the physical constraints of memory and runtime, which we ablated in Section 5.6 of the paper. For example, 25-shot ICL is actually the maximum number of vanilla ICL shots that can be run on a single 48GB A6000 GPU for Qwen-VL. We demonstrate the degradation of increasing numbers of multimodal token-space ICL shots (VizWiz-QwenVL) in Table 5a.

**Effect of shot quality on MTV:**. We assess the connection between textual and activation-space shot quality by comparing MTV using random selection with MTV using high-quality shots selected with the Facility Location algorithm [70]. We apply MTV to QwenVL and use the Qwen GTE embedding model to obtain embeddings for the Facility Location algorithm and present the results in Table 5b. Excitingly, we find that high-quality shots do indeed lead to significant improvements in MTV performance.

**MTV with noisy exemplars.**. We compare the robustness of MTV compared to that of vanilla ICL. For QwenVL on VizWiz and OKVQA, we replace 1 of the 4 examples in each iteration of 4-shot ICL and 4-shot-100-iteration MTV with an example from the opposite dataset. We report both accuracy and degradation in Table 5c

Table 5: **ICL Degradation, Shot-Quality Impact, and Stability** (Left) Degradation of ICL with increasing number of shots. (Right) Impact of shot quality on MTV and stability of ICL vs MTV with noisy examples.

(a) ICL Degradation with Increasing Shots

| ICL Shots | Acc. | % Acc. Increase |
|-----------|------|-----------------|
| 0         | 35.2 | -               |
| 4         | 42.0 | 6.8             |
| 8         | 44.3 | 2.3             |
| 16        | 46.9 | 2.6             |
| 20        | 49.0 | 2.1             |
| 25        | 49.8 | 0.8             |

(b) MTV with High-Quality Shots

| Model            | VizWiz |
|------------------|--------|
| QwenVL-7B        | 35.2   |
| + MTV            | 45.6   |
| + MTV + F.L. Shots | 58.1 |

(c) Stability of ICL vs MTV using QwenVL

|           | VizWiz     | OK-VQA     |
|-----------|------------|------------|
| 4-shot ICL | 41.0 (-1.0) | 61.5 (-0.5) |
| MTV       | 43.4 (-2.2) | 61.9 (-0.1) |

**Attention head generalization on object classification tasks Table 6a**. We also test generalization for object classification tasks identical to the formulation described in Section 5.3. For clarity, MTV shows another kind of generalization when it is leveraged alongside additional explicit ICL samples. This capability is described in Section 5.2. To summarize our experiment, we calculate MTV using the Flowers dataset using 1-shot ICL example for 100 iterations for both the mean activations $\mu_j^{\text{MTV}}$ and the attention head locations $\lambda_j^{\text{MTV}}$. Then, we apply MTV to the CUB task *using the same set of attention head locations from Flowers*. We just calculate the mean activations for the CUB dataset using a 1-shot for 100 iterations (halving our data requirement for this specific scenario). Once again, we find that the heads of MTV can indeed generalize between similar classes.

Table 6: **Generalization & Direct ICL Comparison** (Left) MTV-Flowers evaluated on OK-VQA. (Right) Direct comparison of MTV extracted from 4-shots, 1-iteration (MTV_4shot_1it) compared to 4-shot ICL

(a) Attention Head Generalization

| Model | Flowers | CUB |
|---|---|---|
| ViLA-1.5-8B | | |
| + 1-shot-ICL | 87.4 | 88.4 |
| + **MTV-Flowers**+1-shot-ICL | 89.3 | **89.9** |

(b) Comparison to Other Methods

| Model | VizWiz | OK-VQA |
|---|---|---|
| ViLA-1.5-8B | 28.0 | 32.8 |
| + 4-shot-ICL | 39.3 | 35.6 |
| + **MTV**_4shot_1it | 57.4 | 40.0 |

**MTV one-to-one comparison with ICL - Table 6b**. Although not directly comparable, we consider an extreme case of MTV where we encode only 4-shots of ICL examples for 1 iteration. This matches the exact setting used in standard 4-shot ICL. Interestingly, MTV applied to both VizWiz and OK-VQA exceeds performance on the 4-shot-ICL case and even MTV formulated on 4-shots per 100 iterations for calculating the mean activations. This result suggests that there may be scope for MTV to be effective in both high and low-data regimens. More research needs to be done to explore this idea.

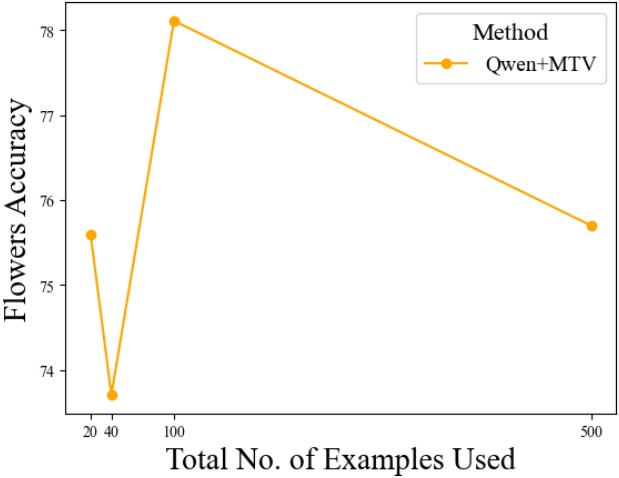

Figure 4: **Efficiency.** We show that for Flowers, MTV does scale to but only up to 100 examples in our experiments.

**Scaling on Flowers Dataset**. We provide additional results on the scaling property of MTV on the Flowers dataset. We again note that the examples are *2-way*, one-shot examples with 2 examples (one positive and one negative) for each sample. As in the main paper, we fix 1 shot per iteration to calculate the mean activations, scaling up to 500 total examples used. Our results show that there is a saturation of MTV at 100 examples (i.e., 1 example per 100 iterations). While this still indicates some scaling as the result is an improvement over 20 examples, the results show that the task vector can reach its best accuracy with fewer shots depending on the complexity of the task. Future work to probe more deeply into the scaling nature of MTV across different tasks would be valuable.

**MTV at extreme shot counts**. We delve further into the scaling capabilities by evaluating the performance of MTV at the maximum number of VizWiz shots per iteration allowable by the memory constraints of a single NVIDIA RTX A6000. The experiment shown in Table 7a indicates that while MTV does continue to scale, there is also certainly a saturation point for VizWiz. The exact saturation point likely depends on the specific task.

**Effect of permutation**. We consider applying five random seeds to different configurations of MTV comparing its variability under permutation of example order to standard few-shot ICL. We present the mean and standard deviation for these experiments in Table 7b. Although not significant, in both 4-shot and 8-shot settings, MTV shows less variability to example permutation. This intuitively makes sense as more examples are averaged over multiple iterations, leading to more stable performance across different seeds.

Table 7: **QwenVL-7B: Generalization & Variability to Permutation** (Left) Evaluation of MTV on extreme shot-iteration counts. (Right) Variability to permutation across 5 seeds.

(a) QwenVL-7B: Extreme shot-count performance

| Model | Accuracy (%) |
|---|---|
| QwenVL-7B | |
| + MTV_20shot_100it | 54.9 |
| + MTV_20shot_200it | 55.1 |
| + MTV_25shot_100it | 56.4 |
| + MTV_25shot_200it | 51.4 |

(b) Permutation Variability

| Model | Accuracy (%) |
|---|---|
| QwenVL-7B | |
| + MTV_4-shot_100it | 45.2 (0.7) |
| + MTV_4-shot_200it | 48.3 (0.4) |
| + MTV_8-shot_100it | 50.4 (0.9) |
| + MTV_8-shot_200it | 51.8 (0.6) |
| + 4-shot ICL | 41.3 (0.8) |
| + 8-shot ICL | 42.9 (1.5) |

**MTV for language-only tasks**. While we show the importance of MTV especially for vision-language tasks, the methodology can be a powerful way to learn tasks in the language-only domain as well. We demonstrate in Table 8a the effectiveness of MTV on two common LLM tasks using LLaMA-3-8B [55]:

Table 8: **Performance on Language and Document Tasks** (Left) Evaluation on English-Spanish and Antonym Generation tasks. (Right) MTV performance across different shot settings on document tasks.

(a) Performance on Language-Only Tasks

| | English-Spanish | Antonym Generation |
|---|---|---|
| 10 shot | 65.2 | 56.0 |
| 400 shot | 68.5 | 57.6 |
| **MTV 4-100** | **76.7** | **61.7** |

(b) MTV Document Task Performance

| | 0-shot | 4-shot | 8-shot | MTV |
|---|---|---|---|---|
| ChartQA | 19.1 | 25.0 | 26.4 | 34.9 |
| TextVQA | 42.4 | 45.4 | 47.1 | 51.0 |

**MTV on additional document datasets**. Multimodal documents are a form of data with complex compositions of visual and textual modalities, with interleaved language, photograph, and chart information. As such, we provide some preliminary results on the effectiveness of MTV on these types of datasets in Table 8b. These encouraging results prompt future research into the domain of leveraging task vectors for learning challenging document tasks.

Here we provide some additional method details about MTV, Visual Task Vectors (VTV) [27], and Function Vectors [81] (FV).

## A.2 MTV-EXTRACT

We describe the particulars of our MTV-EXTRACT algorithm for finding the set of attention head locations that best align with the downstream task as follows ($Q_s$ and $R_s$ are formatted identically to the downstream task):

**Algorithm 1** MTV-EXTRACT for finding task vector locations

---

**Require:** $F$ (LMM), $S$ (examples), $\mu_j$ (mean activations), $Q_s, R_s$ (queries and responses)
**Ensure:** $\lambda_j^{\text{MTV}}$ (optimized attention head locations)
 1: Initialize $\theta$ randomly
 2: **for** $s \leftarrow 1$ to $S$ **do**
 3:     **for** $i \leftarrow 1$ to 32 **do**                              ▷ Sampling heads 32 times
 4:         Sample $\lambda_i \sim \text{Bernoulli}(\sigma(\theta))$
 5:         Replace activations for $\lambda_i$ in $F$ with $\mu_{l,j}$
 6:         Compute output logits $O_s \leftarrow F(Q_s)$            ▷ Pass $Q_s$ to LMM $F$
 7:         $L_i \leftarrow \text{Negative Cross-Entropy}(O_s, R_s)$
 8:     **end for**
 9:     $\theta \leftarrow \text{Adam}(\theta, \nabla_\theta \frac{1}{32} \sum_{i=1}^{32} L_i)$                 ▷ Update rule
10: **end for**
11: Sample final $\lambda_j^{\text{MTV}} \sim \text{Bernoulli}(\sigma(\theta))$         ▷ Final set of head locations
12: **return** $\lambda_j^{\text{MTV}}$

---

We point out a few important factors. It is important to note that none of the parameters of $F$ are being finetuned through any gradient update. We take the negative cross-entropy (negative as MTV_EXTRACT draws inspiration from REINFORCE [91], which is a policy optimization algorithm) between the output logits $O_s$ and the first token of the target response $R_s$ for a simple update scheme. This along with the choice of 32 samples of the Bernoulli distribution are ones we encourage more experimentation with in future work.

### A.3   Visual Task Vectors (VTV) Adaptation for Multimodal ICL

Visual Task Vectors (VTV) [27] were originally designed to be applied to large vision-transformer-based models. We make as few changes as possible to apply this method for multimodal tasks. We preserve VTVs distinct factors like a the usage of 1-shot examples for both calculation of the mean activations and attention head locations regardless of the format of the downstream task. Furthermore, we fix the number of iterations for both mean activation and attention head calculation at 10. Finally, we replace the proposed MSE loss with a cross-entropy loss that is more suited for an LMM task.

### A.4   Function Vectors (FV)

Because Function Vectors describe text-only task vectors, we follow the implementation of Function Vectors [81] almost exactly as LLMs and LMMs are similar. The only major change made is the use of many-shot multimodal ICL examples for mean activation calculation. We preserve the lack of an optimization method for the layer used to replace the mean activations. Rather than performing a standard grid search over the set of layers, we set the layer number to 20 as recommended for LLaMA and LLaMA-based models by the paper. The only other difference is the encoding of multimodal ICL examples. Again due the the similarity between LMMs and text-only LLMs, these tests can be used as needed as long as the multimodal inputs are properly processed by the LMM.

## B   Additional Implementation Details

To run all of our experiments, we use 1 NVIDIA RTX 6000 GPU. Importantly, this includes the runtime and efficiency ablations, which were evaluated on the same GPU for consistency. Please refer to the respective model's paper for their specific implementation details of the architecture. Besides the output token generation length, which varies depending on the standard setting for each task, we use the default generation parameters (e.g. temperature and no. of beams in beam search) recommended for each model. In the following sections, we describe some of the finer nuances of our MTV-EXTRACT process as well as our implementations of the Visual Task Vectors (VTV) and Function Vectors (FV) implementations.

### B.1 VizWiz

**Dataset**. The VizWiz dataset is designed to challenge and evaluate the capabilities of Large Multimodal Models (LMMs) in understanding and responding to real-world visual questions. This dataset is comprised of images accompanied by spoken questions, which have been transcribed and paired with answers. Each image in this dataset is sourced from visually impaired individuals seeking assistance, thereby incorporating a wide array of everyday challenges they face. This setup is inherently diverse and often requires high-level visual understanding combined with contextual reasoning, making them a robust benchmark for assessing the practical utility of LMMs in assistive technologies. The format of the dataset samples is an image paired with a text question. The LMM is required to provide a short response limited to 10 tokens or respond with "unanswerable" if the question is not answerable give the image.

For this research paper, we specifically utilize the VizWiz dataset to benchmark the performance of our proposed task vectors in multimodal in-context learning (MM-ICL) on a dataset that challenges visual scene understanding of LMMs. We extract MTV on the training set and evaluate on the evaluation set containing 4,319 validation image/question pairs.

**Inference details**. We use the standard VQA question-answer response format that is outlined in the QwenVL repository `https://github.com/QwenLM/Qwen-VL`. Put simply, the LMM is presented with an image and a corresponding text question. The response is then expected in a short text format of no more than 10 tokens (set as the "max_tokens" parameter in the LMM). One nuance is the special answer "unanswerable". We handle this by providing MTV and all baselines with the following prompt for every question: "First carefully understand the given examples. Then use the given image and answer the question in the same way as the examples. If the question can not be answered, respond unanswerable. " The official dataset can be downloaded at `https://vizwiz.org/tasks-and-datasets/vqa/`.

### B.2 OK-VQA

**Dataset**. The OK-VQA dataset, differs from traditional VQA datasets in its focus on necessitating knowledge beyond what is presented in the given images. This dataset encompasses over 14,000 questions that are not merely reliant on visual cues but require associative reasoning with external data sources, making it a unique tool for evaluating AI's capability in handling complex, knowledge-driven queries. Thus, we evaluate on this dataset to test whether MTV can be beneficial for this type of reasoning.

We once again extract MTV on the train set and evaluate on the validation set. OK-VQA is formatted as an image with a corresponding text question. However, it is important to note that the text question heavily relies on external knowledge to answer. Examples of questions can be found in Section C.

**Inference details**. We use the standard VQA question-answer response format that is outlined in the QwenVL repository `https://github.com/QwenLM/Qwen-VL`. Put simply, the LMM is presented with an image and a corresponding text question. The response is then expected in a short text format of no more than 10 tokens (set as the "max_tokens" parameter in the LMM). We do not add any additional prompts or special tokens apart from prompt format or image tokens required by the model being evaluated. The official dataset can be downloaded at `https://okvqa.allenai.org/`.

### B.3 Flowers

**Dataset**. Flowers [59] is an object classification dataset that requires fine-grained classification of 102 different flower species. The Flowers dataset is formulated as a 2-way, 1-shot task where one example is the positive sample and the other is the negative sample. In this way, the data poses a unique challenge for MTV having to store examples with two associated images. Thus, given the 2-way examples and the query image, the LMM is tasked with selecting the correct class from the given two options. Examples can be found in Section C

**Implementation Details**. We use the official data released by the authors which is available at `https://www.robots.ox.ac.uk/~vgg/data/flowers/`. We provide a Python code snippet below showing the Flowers data format:

```python
def format_flower(cur_data):
```

```
        pos = cur_data["pos"]
        neg = cur_data["neg"]
        pos_label = cur_data["pos_label"]
        neg_label = cur_data["neg_label"]
        query = cur_data["query"]
        rand_num = random.randint(0,1)
        if rand_num == 0:
            pos_example = f"{pos}</img>What is the type of flower in the image? A.{
                pos_label} B.{neg_label}\nAnswer with the option's letter from the given
                 choice directly. Answer: A\n"

            neg_example = f"{neg}</img>What is the type of flower in the image? A.{
                pos_label} B.{neg_label}\nAnswer with the option's letter from the given
                 choice directly. Answer: B\n"

            cur_query = f"{query}</img>What is the type of flower in the image? A.{
                pos_label} B.{neg_label}\nAnswer with the option's letter from the given
                 choice directly. Answer:"
            query_label = "A"
            return pos_example + neg_example + cur_query, query_label, -1

        else:
            pos_example = f"{pos}</img>What is the type of flower in the image? A.{
                neg_label} B.{pos_label}\nAnswer with the option's letter from the given
                 choice directly. Answer: B\n"

            neg_example = f"{neg}</img>What is the type of flower in the image? A.{
                neg_label} B.{pos_label}\nAnswer with the option's letter from the given
                 choice directly. Answer: A\n"

            cur_query = f"{query}</img>What is the type of flower in the image? A.{
                neg_label} B.{pos_label}\nAnswer with the option's letter from the given
                 choice directly. Answer:"
            query_label = "B"
            return neg_example + pos_example + cur_query, query_label, -1
```

## B.4   CUB

**Dataset**. CUB [82] or CUB-200-2011 is an object classification dataset that tests the fine-grained classification of 200 classes of birds. Similar to the Flowers dataset, CUB is formulated as a 2-way, 1-shot task where one example is the positive sample and the other is the negative sample. In this way, the data poses a unique challenge for MTV having to store examples with two associated images. Thus, given the 2-way examples and the query image, the LMM is tasked with selecting the correct class from the given two options.

**Implementation Details**. We use the official data released by the authors which is available at `https://www.vision.caltech.edu/datasets/cub_200_2011/`. We provide a Python code snippet below showing the Flowers data format:

```
def format_cub(cur_data):
    pos = cur_data["pos"]
    neg = cur_data["neg"]
    pos_label = cur_data["pos_label"]
    neg_label = cur_data["neg_label"]
    query = cur_data["query"]
    rand_num = random.randint(0,1)
    if rand_num == 0:
        pos_example = f"{pos}</img>What is the type of bird in the image? A.{
            pos_label} B.{neg_label}\nAnswer with the option's letter from the given
             choice directly. Answer: A\n"

        neg_example = f"{neg}</img>What is the type of bird in the image? A.{
            pos_label} B.{neg_label}\nAnswer with the option's letter from the given
             choice directly. Answer: B\n"
```

```
        cur_query = f"{query}</img>What is the type of bird in the image? A.{
            pos_label} B.{neg_label}\nAnswer with the option's letter from the given
             choice directly. Answer:"
        query_label = "A"
        return pos_example + neg_example + cur_query, query_label, -1

    else:
        pos_example = f"{pos}</img>What is the type of bird in the image? A.{
            neg_label} B.{pos_label}\nAnswer with the option's letter from the given
             choice directly. Answer: B\n"

        neg_example = f"{neg}</img>What is the type of bird in the image? A.{
            neg_label} B.{pos_label}\nAnswer with the option's letter from the given
             choice directly. Answer: A\n"

        cur_query = f"{query}</img>What is the type of bird in the image? A.{
            neg_label} B.{pos_label}\nAnswer with the option's letter from the given
             choice directly. Answer:"
        query_label = "B"
        return neg_example + pos_example + cur_query, query_label, -1
```

## C  Qualitative Visualizations

We present further qualitative success and failure cases of **QwenVL-MTV** in Figure 5 on OK-VQA
and Flowers.

## D  Licenses and Privacy

The license, PII, and consent details of each dataset are in the respective papers. In addition, we wish
to emphasize that the datasets we use do not contain any harmful or offensive content, as many other
papers in the field also use them. Thus, we do not anticipate a specific negative impact, but, as with
any machine learning method, we recommend exercising caution.

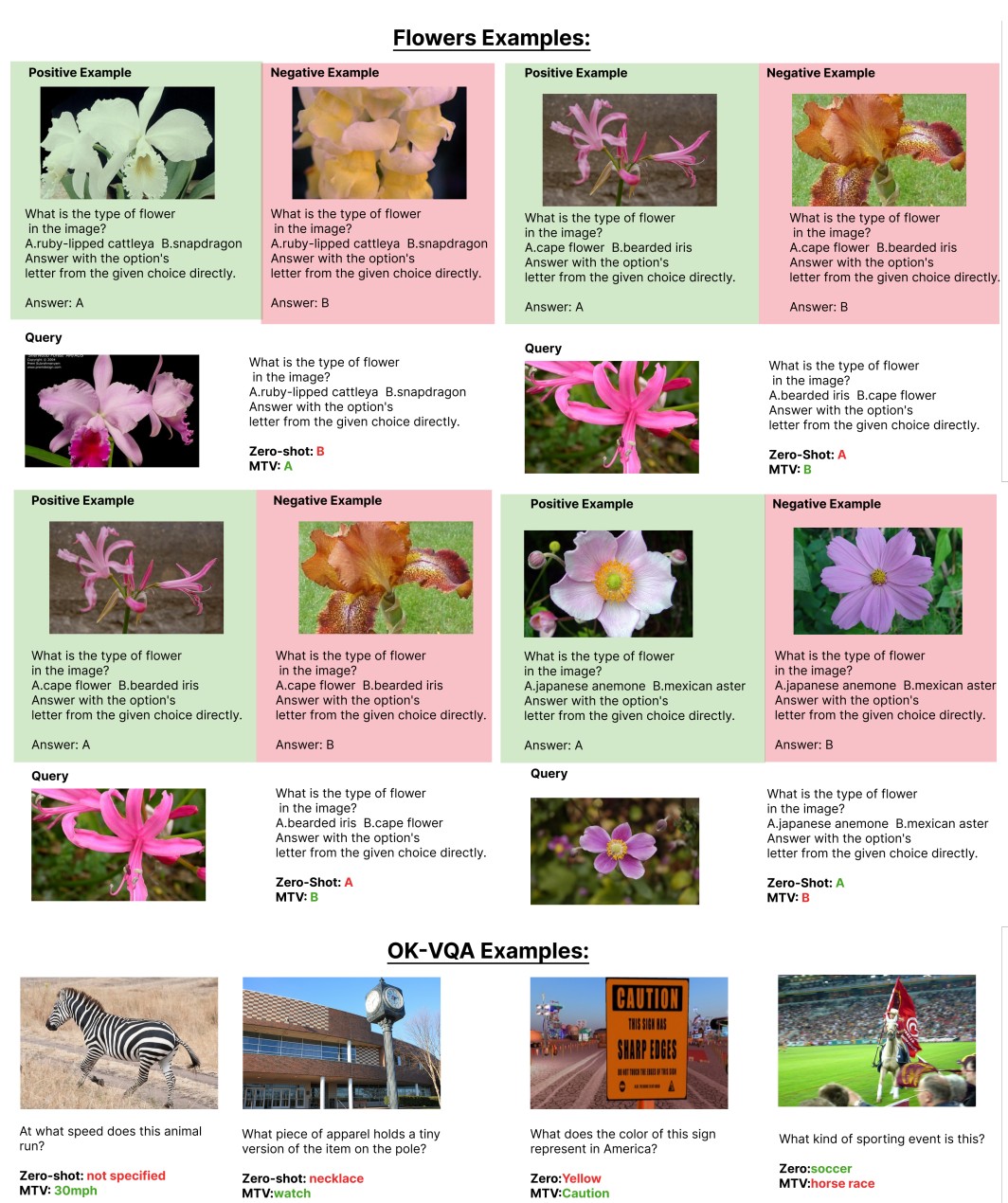

Figure 5: **Efficiency.** We show that for Flowers, MTV does scale to but only up to 100 examples in our experiments.

