# OpenReview forum: "Multimodal Task Vectors Enable Many-Shot Multimodal In-Context Learning"
_NeurIPS.cc/2024/Conference — NeurIPS 2024 poster_

### Official Review · Reviewer_Xn5t · 2024-06-13

**Soundness:** 3
**Presentation:** 3
**Contribution:** 3
**Rating:** 7
**Confidence:** 4

**Summary:**

This work presents MTV to enhance the in-context learning (ICL) capabilities of LMMs, which typically have limited context length, especially when dealing with multimodal data that includes both text and images. MTV addresses this by compressing many-shot examples into compact implicit representations within the model's attention heads. The approach involves calculating mean activations of attention heads across multiple inference iterations and selecting optimal locations for these activations using an adapted REINFORCE algorithm. This method allows LMMs to effectively use more examples than their context length permits, improving performance on various vision-and-language tasks without finetuning. Experiments demonstrate that MTV not only scales with more examples but also generalizes to similar out-of-domain tasks and works alongside explicit few-shot examples, offering a practical solution for efficient and effective many-shot multimodal ICL.

**Strengths:**

1. Interesting idea and finding, MTV is an efficient and lightweight method to use training data and off-the-shelf LMMs.
2. The writing is well-written and easy-to-follow overall.
3. The results seem good, models are evaluated on two tasks, four benchmarks and three models. In addition, the efficiency is also evaluated. I like the efficiency evaluation.

**Weaknesses:**

1. Minor: the dataset can always be included more: Try MMMU[1], OpenFlamingo and Otter did worth with 1/3/5-shot examples.
2. Minor: the models can also be included more: Try Mantis[2]

[1] Yue et al., MMMU: A Massive Multi-discipline Multimodal Understanding and Reasoning Benchmark for Expert AGI. 2023 ArXiv.

[2] Jiang et al., Mantis: Interleaved Multi-Image Instruction Tuning, 2024 ArXiv.

**Questions:**

1. Can this method work on LMMs that can only take in one image? Because most of models currently being evaluated are models that can take multiple images. Since your work is like to use the in-context examples as the prior, then I think it's possible.

**Limitations:**

No significant limitations

---

> ### Author Rebuttal · Authors · 2024-08-06
>
> We thank the reviewer for their valuable comments. In the following, we provide a response to the questions raised in the review:
>
> **Additional MANTIS results.** As suggested by the reviewer, we present some additional results of our method using the MANTIS-LLama3 model on OKVQA, Vizwiz, Flower, and CUB:
>
> |       | OK-VQA | VizWiz | Flower | CUB  |
> |-------|-------|--------|--------|------|
> | Zero-Shot  | 51.7  | 36.3   | -      | -    |
> | 4-shot ICL   | 52.5  | 26.4   | 88.7   | 84.0 |
> | MTV-100   | 52.8  | 51.0   | 89.8   | 89.7 |
>
> We will include these results in the final draft of the paper.
>
> **Additional Dataset results.** As suggested by the reviewer, we present additional results of our method on the following benchmarks using VILA-1.5:
>
> |          | 0-shot   | 4-shot   | 8-shot   | MTV  |
> |----------|-----|-----|-----|------|
> | ChartQA  | 19.1| 25  | 26.4| 34.9 |
> | TextVQA  | 42.4| 45.4| 47.1| 51   |
>
> **Leveraging MTV for Single-Image LMMs.** We thank the author for the suggestion to leverage MTV for single-image LMMs. This question is especially interesting as single-image pretraining causes the underlying model to struggle with handling multi-image inputs [1].
>
> We apply MTV to LLaVA-1.5-13B on VizWiz. The model has a zero-shot accuracy of 54.5% and an accuracy of *62.1% with MTV*. This result is encouraging as it indicates MTV’s ability to enable multi-image, few-shot, and many-shot multimodal reasoning for models pretrained on exclusively single-image data.
>
> We hope the above points have clarified and addressed your concerns. We would be happy to provide any further clarifications as requested.
>
> References:
>
> [1] Doveh, S., Perek, S., Mirza, M.J., Alfassy, A., Arbelle, A., Ullman, S., & Karlinsky, L. (2024). Towards Multimodal In-Context Learning for Vision & Language Models. ArXiv, abs/2403.12736.

---

> > ### Comment · Reviewer_Xn5t · 2024-08-07
> >
> > Thanks for the addtional results, hope to see these results in the final version.
> > I raised my score as an outcome of the discussion.

---

### Official Review · Reviewer_p88K · 2024-07-13

**Soundness:** 1
**Presentation:** 1
**Contribution:** 1
**Rating:** 1
**Confidence:** 5

**Summary:**

This paper should be desk rejected as the single PDF submission does not have the paper checklist.
It would be unfair to make an exception as this requirement has been clearly stated at https://neurips.cc/Conferences/2024/CallForPapers and the latex template file.

**Strengths:**

NA

**Weaknesses:**

NA

**Questions:**

NA

---

> ### Author Rebuttal · Authors · 2024-08-06
>
> Please note that our checklist is provided in our Supplementary pdf, and thus, the PCs have decided not to desk-reject this paper. As this has been universally applied to all submissions, our submission is thus in accordance with guidelines, and fairness should not be adversely affected.

---

### Official Review · Reviewer_3gQC · 2024-07-13

**Soundness:** 2
**Presentation:** 2
**Contribution:** 3
**Rating:** 5
**Confidence:** 4

**Summary:**

In the context of multimodal understanding, this work studies the area of many shot, long-context ICL (in context-learning) in natively multimodal models (large multimodal models i.e. LMMs, where images and text are interleaved). The premise proposed is that the pretrain time context of existing LMMs is prohibitive w.r.t generalization to longer contexts for adding many-shot multimodal examples. To leverage many examples under this context budget, this work proposes a technique to compress i.e. encode these multimodal exemplars into the weight space (multimodal task vectors i.e. MTV) by encoding them via mean attention head activations and replacing these mean activations into heads which are most aligned for a downstream task (which is potentially out of domain / zero shot). This selection is done at inference time via a REINFORCE based selection mechanism. Empirically, consistent improvements are demonstrated compared to the ICL baseline across open models (where weights can be accessed) on 4 tasks covering visual question answering and object classification.

**Strengths:**

This paper addresses and important and relevant area of multimodal Q/A or understanding in natively multimodal models, where not many benchmarks exist to understand the long context multimodal capabilities of LMMs. I like the coherence in the methodology and the stepwise presentation in the paper. Figure 1 gives a concise overview of the method being proposed.

**Weaknesses:**

While the empirical improvements are encouraging, following are some weaknesses where I have concerns regarding the motivation of the method and it would be great to have a discussion on these:

1. How is the proposed method uniquely situated in the context of "multimodal" QnA? The context length limitation, encoding exemplars efficiently and tuning head selection for a downstream task seem to be applicable in the text-only domain too. Can this same technique be applied to the text-only space as well? What is specific about the methodology proposed which would make sense for the multimodal domain but probably not work for text?

2. There are repeated claims on images being "more expensive" in the token space e.g. “Image embeddings are expensive”, “the images require more tokens to embed”. I believe these claims could be concretized better by doing a simple analysis of difference in tokens, because I believe this difference is usually not prohibitive. It would also be nice to have numbers on the context lengths in the benchmarks mentioned.

3. There is no evidence of degradation of model performance with increasing context length in the number of multimodal shots. Without this, it is hard to understand the motivation for choosing to encode shots in the weight space. If one can see the degradation being more rapid for vanilla ICL compared to this method, the motivation would be much more convincing.

3. Why does one need REINFORCE based selection? Concretely, how would the current method compare with the baseline of simply replacing all heads with the mean activations? It is unclear why the authors chose to use REINFORCE.

4. Having many shots in the context (v/s in the weight space) provides the benefit of interpretability - what is the intuition on losing on this benefit? Does shot quality in the textual space correspond to shot quality in the weight space? As a follow up, how does the "cost of prompt iteration" compare between vanilla ICL v/s MTV?

5. How robust is this technique to shot quality e.g. noisy exemplars? Is it more or less robust than vanilla ICL?

6. The empirical results focus on comparisons with vanilla ICL. While encouraging, it would be convincing if the method shows performance improvements on long context tasks where vanilla ICL fails but this method works. Further, the tasks considered are still limited and a wider range of longer-context multimodal benchmarks could be considered. (to name some e.g. MMMU, TextVQA, DocVQA).

**Questions:**

Besides the above, here's a suggestion: there are many abbreviations (e.g. VL benchmarks) which could be expanded clearly.

**Limitations:**

Limitations are mentioned.

---

> ### Author Rebuttal · Authors · 2024-08-06
>
> We thank the reviewer for the insightful comments and respond to all points in the following, incorporating all clarifications and additional results into the final paper:
>
> **MTVs for Text-only domain.** Using LLaMA3, we evaluate MTV on text-only tasks. The two tasks are English-Spanish translation and Antonym generation, as done in [1]:
>
> |             | English-Spanish | Antonym Generation |
> |-------------|-----------------|---------|
> | 10 shot     | 65.2            | 56.0    |
> | 400 shot    | 68.5            | 57.6    |
> | MTV 4-100   | 76.7            | 61.7    |
>
> We find that MTV is also effective in the text-only domain.
>
> **MTV’s unique fit for multimodal tasks.** As suggested by the reviewer, we provide the token count per image for each model to highlight the cost of image inputs:
>
> | Model Name | Per Image Token Length | Total Context Length |
> |------------|-------------------|----------------------|
> | VILA       | 144               | 8192                 |
> | Idefics2   | 64                | 8192                 |
> | QwenVL     | 256               | 8192                 |
>
> A maximum of 8192/64 = 128 images can be encoded assuming no language inputs, already lower than the number in base MTV (400 shots).
>
> Considering language tokens, we present the percentage of the input that is text and image on VILA 1.5 for datasets we evaluated on:
>
> | Dataset | % Input Text | % Input Image |
> |-------|----------|------------|
> | VizWiz     |  6.6   |    93.4  |
> | OK-VQA     |      8.4 |    91.6   |
> | Flowers     |   12.7       |     87.3    |
> | CUB  |  14        |    86      |
>
> This shows that images quickly consume many of the input tokens for multimodal tasks. The data presented thus demonstrates the value of MTV, especially in the multimodal domain.
>
> **Motivation for encoding shots in the activation space.** We highlight our paper’s motivation in addressing the context length token limitation of LMMs by encoding ICL shots in the activation space. An additional limiting factor in the token space is the physical constraints of memory and runtime, which we ablated in Section 5.6 of the paper. For example, 25-shot ICL is actually the maximum number of vanilla ICL shots that can be run on a single 48GB A6000 GPU for Qwen-VL.
> Regardless, following the reviewer’s suggestion, we demonstrate the degradation of increasing numbers of multimodal token-space ICL shots (VizWiz-QwenVL):
>
> | ICL Shots | Accuracy | % Accuracy Increase |
> |-------|----------|------------|
> | 0     |  35.2    |    -     |
> | 4     |   42.0     |    6.8     |
> | 8     |    44.3      |    2.3     |
> | 16    |   46.9       |      2.6      |
> | 20    |      49.0    |    2.1        |
> | 25    |      49.8    |    0.8       |
>
> Compared to the results shown in Figure 2 of our paper, vanilla ICL improvement degrades at lower accuracy than MTV.
>
> **REINFORCE-based attention-head extraction.** We perform an experiment replacing mean activations of ALL heads for VizWiz. Interestingly, QwenVL achieves 0% accuracy on Vizwiz in this setting. We assume this is due to the query contexts being completely overwritten by the mean activations.
>
> Past work has shown that some subset of the heads include important ICL information and showed several ways to extract this information [1,2,3]. In our work, we select this subset of attention heads using a REINFORCE-based method to optimize a distribution over a *set* of attention heads as opposed to one-at-a-time like CMA [1], leading to superior performance (Section 5.5).
>
> **MTV Shot Quality.**  We assess the connection between textual and activation-space shot quality by comparing MTV using random selection with MTV using high-quality shots selected with the Facility Location algorithm [5]. We apply MTV to QwenVL and use the Qwen GTE embedding model to obtain embeddings for the Facility Location algorithm:
>
> | Model            | VizWiz |
> |--------------------------|---------|
> | QwenVL-7B         | 35.2   |
> | + MTV                    | 45.6 |
> | + MTV + F.L. Shots| 58.1   |
>
> We find that choosing higher-quality examples drastically improves the performance of MTV, indicating that shot quality in the textual and activation spaces is closely connected.
>
> **MTV interpretability.** We appreciate the reviewer's note on interpretability. We posit an alternative perspective that MTV shots are no less interpretable but are simply the same set of examples represented as a set of activations. Furthermore, task vector methods like MTV give additional insight into which attention heads are relevant for an ICL task [1,4].
>
> **Cost of prompt iteration.** Please note that we provide an analysis of the cost of prompt iteration in Section 5.6 of the paper. We show that MTV is more runtime efficient than 4-shot ICL and more memory efficient than 16-shot ICL.
>
> **MTV with noisy exemplars.** We compare the robustness of MTV with vanilla ICL as suggested. For QwenVL on VizWiz and OKVQA, we replace 1 of the 4 examples in each iteration of 4-shot ICL and 4-shot-100-iteration MTV with an example from the opposite dataset. We report both accuracy and degradation.
>
> |        | Qwen VizWiz | Qwen OK-VQA |
> |--------|-------------|-------------|
> | 4-shot | 41.0 (-1.0) | 61.5 (-0.5) |
> | MTV    | 43.4 (-2.2) | 61.9 (-0.1) |
>
> The results suggest that MTV and vanilla ICL have similar robustness to noisy examples.
>
> **Document datasets.** We evaluated MTV on document datasets ChartQA and TextVQA with VILA-1.5:
>
> |          | 0-shot  | 4-shot  | 8-shot  | MTV  |
> |----------|-----|-----|-----|------|
> | ChartQA  | 19.1| 25  | 26.4| 34.9 |
> | TextVQA  | 42.4| 45.4| 47.1| 51   |
>
> These results indicate MTV’s capabilities on document-based tasks.
>
> Thank you. We hope this addresses all concerns, and we are happy to clarify anything further.
>
> References:
>
> [1] Todd et. al. FV
>
> [2] Hendel et. al. "In-Context Learning Creates Task Vectors."
>
> [3] Olsson et. al. "In-context Learning and Induction Heads."
>
> [4] Hojel. et. al. VTV.
>
> [5] Schreiber et. al. Apricot.

---

> > ### Comment · Reviewer_3gQC · 2024-08-10
> >
> > Thank you for the detailed responses. After going through the comments and other reviewer's comments, I have decided to raise my score. I look forward to seeing these results and the surrounding discussion in the final version.

---

### Official Review · Reviewer_dwXj · 2024-07-25

**Soundness:** 3
**Presentation:** 2
**Contribution:** 3
**Rating:** 5
**Confidence:** 3

**Summary:**

In-context learning with many examples can be effective for learning new tasks. However, there are challenges with many-shot multimodal in-context learning (ICL), such as the limitation caused by the model’s context length. This issue is more challenging in the multimodal setting because it processes both images and text, requiring additional tokens. Therefore, a method to compress this information into fewer tokens without finetuning is necessary. The paper proposes using large language models (LLMs) to perform many-shot in-context learning with multimodal task vectors (MTVs). MTVs are compact, implicit representations of in-context samples, compressed by the model’s attention head. The paper provides experiments showing that outperform existing methods.

**Strengths:**

The paper is generally well-written.

The motivation is clear, and the research question, and challenging and up-to-date.

The paper provides experiments showing that the proposed method outperforms existing methods.

The paper also provides ablation studies to better analyze and understand the model.

**Weaknesses:**

I would recommend the authors compare their work with “Many-Shot In-Context Learning
in Multimodal Foundation Models”. I think this can be a concurrent work, however, a general comparison of the methods and adding the reference to the paper helps the paper.

I am a bit confused with the number of data used in each case in Tables 1 and 2. Can the authors elaborate on that? Also, can the authors explain more about the MTV+ 1-shot ICL setting?
Is the setting for MTV in Table 1 2-way 1-shot always?

While the FV and VTV are other existing methods, the paper does not clearly explain those in the main text and related work. I would recommend adding an extra link to the explanation on the supplementary material. Also, would the authors elaborate on the main differences here as well?

Based on Figure 2, the best shot is 16 and iterations 100. However, the paper does not provide any data on shots more than 16 and iterations more than 100, which makes the conclusion suboptimal. Similarly, the paper discusses the effect of permutation only with one experiment (running for different seeds) which is not enough.



Writing:

In line 227, (3) is extra

Referring to steps 1 and 2 in lines 42-55 while these steps are not clearly defined makes it hard to follow the text.

I would recommend the authors explain more about the notion of interleaved data in the paper. The current writing can be difficult to read.

In section 5.2, the reference to the table is missing

Reference missing on “scaling on flowers dataset” section

References to sections are missing in A1

**Questions:**

please refer to the weaknesses.

**Limitations:**

The paper includes limitations.

---

> ### Author Rebuttal · Authors · 2024-08-06
>
> We thank the reviewer for the insightful comments and respond to all points in the following, incorporating all suggested changes into the paper:
>
> **Comparison with “Many-Shot In-Context Learning in Multimodal Foundation Models”.** We thank the reviewer for sharing this concurrent work with us and will add the reference to our final version. While the topic of many-shot in-context learning is a shared feature with our work, one should note that the suggested paper focuses exclusively on closed, proprietary models such as GPT-4o and Gemini-1.5 for image classification. Our work focuses on enabling many-shot, in-context learning capabilities for open-source, interleaved models with limited context lengths. We also evaluate on VQA benchmarks, not just image classification tasks.
>
> **Tables 1 and 2 format clarification.** We clarify the exact format and number of shots for each experiment. In Table 1, the prompt for zero-shot is formatted as simply ``<Query>`` (the question to be answered). For 4-shot, the prompt consists of four randomly-selected concatenated examples of multimodal question-answer pairs followed by the query: ``<Q1> <A1> <Q2> <A2> <Q3> <A3> <Q4> <A4> <Query>``. The 8-shot case is identical except with 8 exemplars instead of 4.
>
> Finally, we use a setting of MTV with 400 examples (4 shots per 100 iterations) for calculating mean activations and 100 examples (zero-shot per 100 iterations) for extracting the attention-head locations.
>
> For Table 2, the 2-way-1-shot formulation is used for all experiments. The prompt structure includes one positive example, one negative example, and the query image: ``<image_1> <class_label_1> <image_2> <class_label_2> <query_image>``, where the task is to classify the ``<query_image>`` correctly as one of the given classes. The baseline consists of just a single one of these prompts, while the MTV case calculates mean activations from 100 iterations of one 2-way-1-shot example. The exact prompt we use is provided in Section C.3 of the supplementary material.
>
>
> **Function Vectors (FV) and Visual Task Vectors (VTV) methods explained.** We include a more detailed description of Function Vectors (FV) [1] and Visual Task Vectors (VTV) [2]:
>
>
> FV are a type of text-only task vector that has the following key differences from our method:
> Function Vectors uses a fixed number of shots without exploring the variation of both the number of shots or iterations for many-shot ICL.
> FV utilizes Causal Mediation Analysis [3] to extract the attention head locations. At a high level, this method selects attention heads by calculating the causal impact of each attention head (i.e. how much the LLM’s probability of outputting a correct answer increases given a set of corrupted, incorrect, ICL examples). The layer to replace activations is chosen via a simple linear search. This is in contrast to MTV which selects attention heads by learning a distribution over the attention heads across all layers.
>
> VTV is a type of image-only task vector that has the following key difference from our method:
> VTV is a visual prompting method for vision-transformer models [2]
> VTV also uses a fixed number of single-shot examples without exploring the variation of shots and iterations to enable many-shot ICL.
> In contrast to VTV, MTV uses a token-level loss to align with the usage of an LMM for a multimodal task.
> VTV uses a single set of one-shot examples for both mean activation calculation and attention head extraction. In contrast, MTV decouples this process by first calculating the mean activations and then using a separate set of examples that are specifically aligned with the format of the downstream task.
>
> We find that our method’s distinct properties—increase in shots and iterations, token-level loss, and the decoupling of the activation aggregation and attention-head extraction—are key factors that enable MTV to improve over FV and VTV.
>
> **MTV with more shots and iterations.** Following the reviewer’s suggestion, we present results on MTV with higher shots (20, 25) and iteration counts (200, 400) for Qwen on Vizwiz. 25 shots is the highest number allowable by the memory constraints (48 GB) of a single A6000 GPU. We present the results here:
>
> | Shots | Iterations | Accuracy |
> |-------|------------|----------|
> | 20    | 100        | 54.9     |
> | 20    | 200        | 55.1     |
> | 25    | 100        | 56.4     |
> | 25    | 200        | 51.4     |
>
>
> The purpose of our experiment shown in Figure 2 is to show the *ability of MTV to scale*, not necessarily to find the optimal shot-iteration setting. This exact value will differ based on the dataset and model. Nevertheless, the above results demonstrate the ability of MTV to scale even beyond 16 shots per iteration.
>
> **Measuring the impact of permutation.** As the reviewer suggested, we explore the effect of permutation in more depth by evaluating Qwen-VL on VizWiz using the following (shot, iteration) counts for calculating mean activations: (4,100); (4,200); (8,100); (8,200). We evaluate on 5 seeds for each setting as well as 4-shot and 8-shot ICL as a baseline. In the following table, we present the mean and standard deviation for each run:
>
> *MTV:*
>
> | Shots | Iterations | Accuracy |
> |-------|------------|----------|
> | 4    | 100        | 45.2 (.7)  |
> | 4    | 200        | 48.3 (.4)     |
> | 8    | 100        | 50.4 (.9)    |
> | 8    | 200        | 51.8 (.6)    |
>
> *Few-shot*
>
> | Shots | Accuracy |
> |-------|------------|
> | 4    | 41.3 (0.8)   |
> | 8    | 42.9 (1.5)   |
>
>
> MTV for 4 and 8 shots for different iterations is always more stable with respect to example permutation compared to vanilla ICL in our experiments.
>
> Thank you. We hope these points address all of your concerns, and we are happy to clarify anything further. We also commit to fixing all paper writing errors pointed out in the final paper draft.
>
>
> References:
>
> [1] Todd et. al. FV
>
> [2] Hojel et. al. VTV
>
> [3] Imai et. al. "A General Approach to Causal Mediation Analysis."

---

> > ### Comment · Reviewer_dwXj · 2024-08-12
> > **response to the rebuttal**
> >
> > I thank the authors for the rebuttal. After going through the answers, I stick to my current rating.

---

> > > ### Author Response · Authors · 2024-08-12
> > > **Reply To Reviewer**
> > >
> > > We thank the reviewer for taking the time to review our rebuttal. We would be happy to know if there are any remaining concerns that we can clarify or address.

---

### Author Response · Authors · 2024-08-06
**High-level Summary**

We thank the reviewers for their valuable feedback. In this work, we present Multimodal Task Vectors (MTV)--- compact implicit representations of many-shot in-context examples compressed in the model’s attention heads— and leverage them for many-shot in-context learning in LMMs.

We are encouraged that the reviewers find our work and findings to be "important" (``dwXj``), "challenging" (``3gQC``), and "interesting" (``Xn5t``). The reviewers also noted the strength of our MTV method, which "outperforms existing methods" (``dwXj``), as well as the insightful nature of the ablation studies and efficiency results (‘Xn5t’), which help to ‘better analyze and understand the model’ (``dwXj``). Furthermore, they all noted our paper to be "well-written" (``dwXj``) and "coherent" (``3gQC``), especially appreciating the "step-by-step" (``Xn5t``) presentation and the clarity of Figure 1.

We address all concerns separately in the section and look forward to an open and constructive discussion with the reviewers.

---

### Decision · Program_Chairs · 2024-09-25

**Decision:**

Accept (poster)

**Comment:**

This paper received 4 reviews, but the review that gave a score of 1 due to desk reject reasons has been ignored since the case was already handled by the PCs. The paper addresses an open and highly challenging problem in the MLLM space, namely to maximize its potential for many-shot in-context learning. The reviewers find that the paper is well written, with a good motivation, and good results. One reviewer had concerns about how the paper is unique in the large landscape of MLLMs, as well as open questions about design choices. These questions were positively answered in the rebuttal, resulting in an accept consensus amongst the reviewers. The AC agrees with the general applicability and broad use of the method and recommends acceptance.